# Prevalence and Factors Associated with Mental and Emotional Health Outcomes among Africans during the COVID-19 Lockdown Period—A Web-based Cross-Sectional Study

**DOI:** 10.3390/ijerph18030899

**Published:** 2021-01-21

**Authors:** Raymond Langsi, Uchechukwu L Osuagwu, Piwuna Christopher Goson, Emmanuel Kwasi Abu, Khathutshelo P Mashige, Bernadine Ekpenyong, Godwin O Ovenseri-Ogbomo, Timothy Chikasirimobi G, Chundung Asabe Miner, Tanko Ishaya, Richard Oloruntoba, Obinna Nwaeze, Deborah Donald Charwe, Kingsley Emwinyore Agho

**Affiliations:** 1Health Division, University of Bamenda, Bambili P. O. Box 39, Cameroon; raylangsi@yahoo.com; 2Diabetes, Obesity and Metabolism Translational Research Unit, Western Sydney University, Campbelltown, NSW 2560, Australia; 3African Vision Research Institute (AVRI), Discipline of Optometry, University of KwaZulu-Natal, Westville Campus, Durban 3629, South Africa; mashigek@ukzn.ac.za (K.P.M.); bekpenyong@unical.edu.ng (B.E.); godwin.ovenseri-ogbomo@uniben.edu (G.O.O.-O.); k.agho@westernsydney.edu.au (K.E.A.); 4Department of Psychiatry, College of Health Sciences, University of Jos, Jos 930003, Nigeria; piwunag@unijos.edu.ng; 5Department of Optometry and Vision Science, School of Allied Health Sciences, College of Health and Allied Sciences, University of Cape Coast, Cape Coast 00233, Ghana; eabu@ucc.edu.gh; 6Department of Public Health, Faculty of Allied Medical Sciences, College of Medical Sciences, University of Calabar, Cross River State 540271, Nigeria; 7Department of Optometry, College of Applied Medical Sciences, Qassim University, Buraydah 51452, Saudi Arabia; 8Department of Optometry, Faculty of Life Sciences, University of Benin, Benin City 300283, Nigeria; 9Department of Optometry and Vision Sciences, School of public health, Biomedical sciences and technology, Masinde Muliro University of Science and Technology, Kakamega 50100, Kenya; chikasirimobi@gmail.com; 10Department of Community Medicine, College of Health Sciences, University of Jos, Jos 930003, Nigeria; minerc@unijos.edu.ng; 11Department of Computer Science, University of Jos, Jos 930003, Nigeria; ishayat@unijos.edu.ng; 12School of Management and Marketing, Curtin Business School, Curtin University, Bentley, WA 6151, Australia; Richard.Oloruntoba@curtin.edu.au; 13County Durham and Darlington, National Health Service (NHS) Foundation, DL3 0PD, UK; o.nwaeze@nhs.net; 14Tanzania Food and Nutrition Center, Dar es Salaam P.O. Box 977, Tanzania; mischarwe@yahoo.co.uk; 15School of Health Science, Western Sydney University, Campbelltown, NSW 2560, Australia

**Keywords:** COVID-19, sub-Saharan Africa, mental health, feeling anxious, worried, frustrated, psychological problem, bored and angry

## Abstract

Mental health and emotional responses to the effects of COVID-19 lockdown in sub-Saharan Africa (SSA) are of serious public health concern and may negatively affect the mental health status of people. Hence, this study assessed the prevalence of mental health symptoms as well as emotional reactions among sub-Saharan Africans (SSAs) and associated factors among SSAs during the COVID-19 lockdown period. This was a web-based cross-sectional study on mental health and emotional features from 2005 respondents in seven SSA countries. This study was conducted between 17 April and 17 May 2020 corresponding to the lockdown period in most SSA countries. Respondents aged 18 years and above and the self-reported symptoms were feeling anxious, being worried, angry, bored and frustrated. These were the main outcomes and were treated as dichotomous variables. Univariate and multivariate logistic regression analyses were used to identify the factors associated with these symptoms. We found that over half (52.2%) of the participants reported any of the mental health symptoms and the prevalence of feeling bored was 70.5% followed by feeling anxious (59.1%), being worried (57.5%), frustrated (51.5%) and angry (22.3%) during the COVID-19 pandemic. Multivariate analysis revealed that males, those aged >28 years, those who lived in Central and Southern Africa, those who were not married, the unemployed, those living with more than six persons in a household, had higher odds of mental health and emotional symptoms. Similarly, people who perceived low risk of contracting the infection, and those who thought the pandemic would not continue after the lockdown had higher odds of mental health and emotional symptoms. Health care workers had lower odds for feeling angry than non-healthcare workers. During the COVID-19 lockdown periods in SSA, about one in two participants reported mental health and emotional symptoms. Public health measures can be effectively used to identify target groups for prevention and treatment of mental health and emotional symptoms. Such interventions should be an integral component of SSA governments’ response and recovery strategies of any future pandemic.

## 1. Introduction

The outbreak of coronavirus disease is causing considerable acute risk to public health and might also have an unanticipated impact on the mental health symptoms exhibited by people across the globe [1]. At the time of this writing, there were more than 63.8 million confirmed cases and 1.48 million deaths from COVID-19 globally. Africa has recorded more than 2.18 million confirmed cases, and 52,000 people have died from COVID-19. Apart from the personal hygiene practices, the imposition of tight restrictions on movements through lockdowns, most governments also implemented social distancing, self-isolation and quarantine measures worldwide in order to contain community spread of the pandemic. These measures brought about a trail of near economic standstill [2,3] and devastating mental health, emotional and psychosocial consequences [4,5].

The enforcement of strict nationwide lockdown measures disrupted the day-to-day lives of the general public as a result of the closure of businesses, culminating in the shrinkage of the already fragile economies of most of the underdeveloped Sub-Saharan African (SSA) countries [6]. These measures have also resulted in mass unemployment and huge job losses across many countries, especially in SSA, where most citizens are self-employed and live below the poverty line [7]. These uncertainties, and worries relating to finances, job insecurity and access to quality health coverage, have a detrimental impact on mental wellbeing [8]. The insecurity about the future, the ceaseless news coverage and overbearing daily social media messages serve as stressors of feeling anxious resulting in disruptions in sleeping and eating patterns leading to irritability, low motivation, and increased alcohol and drug abuse [9].

Recent surveys report an increase in the cases of post-lockdown anxiety and paranoia, and a general feeling of loss, including loss of income and routine or social interaction [10,11,12]. Previous epidemics have induced widespread fear, loneliness and psychological consequences and COVID-19 is showing similar effects. An evaluation using the experience from previous outbreaks such as the 2003 SARS in China and later four countries, recommend intensification of the ongoing surveillance and monitoring of the psychological consequences for outbreaks of pandemics from life-threatening diseases, establishing early targeted mental health interventions, should become routine as part of preparedness efforts worldwide. There has been about 60% increase in emergency calls related to domestic violence across Europe [13], an increase in domestic abuse incidents of 32–36% in France, 21–35% increase across the USA, and a 25% increase in the UK [9]. A survey carried out by the United Nations Population Fund (UNFPA) for West, and Central Africa revealed a 35% increase in gender-based violence (GBV) and rape cases in West Africa and a 62% affirmation of GBV experiences since the lockdown [14].

Children and adolescents have also had their own challenges, including the closure of schools and universities, resulting in significant disruption to daily routines and leisure, examination postponement and graduation cancellations [15,16]. Closure of recreation facilities, national parks and playgrounds have had the unintended consequences of a reduction in physical activity and increase in sedentary lifestyle and obesity, exposing the population to an increased risk of developing or deteriorating existing chronic health conditions such as diabetes mellitus and hypertension [17]. Furthermore, the United Nations World Tourism Organization (UNWTO) indicates that international tourist arrivals to Africa decreased by 35% between January to April 2020 as a result of the pandemic and therefore countries that are heavily dependent on the expenditure of international tourists have witnessed dwindled injections of tourism-based foreign income, and massive job losses [18]. Together with the confusion caused by the rapid spread of various misinformation about COVID-19 [19], these present a vicious cycle between preventive measures against the virus and increase in the risk factors associated with severe manifestations of COVID-19.

Even though COVID-19 infection rates and related mortality among Africans fall far below the forecasts by the WHO [14], the values were not meeting the prediction at the time of writing. The impact of the disease on mental health symptoms in SSA could be dire, given the region’s weak health care systems and low uptake of mental health services [20]. COVID-19 has not only created mental health disorders with catastrophic emotional changes but has also interrupted essential mental health services just when they were needed most [7]. However, as shown in a rapid review of published articles on mental health services during COVID-19 involving six SSA countries, the authors suggested that efforts to control the disease transmission should be contextualized in the region [6]. This study builds upon the findings from Semo et al. [6] which identified sub-population most affected that could be targeted for future mental health services in the region. As noted elsewhere, mental health services and resources are being delivered through online platforms [17], but the sub-regions low digital literacy penetration makes virtual mental health services a limited preference for service delivery [6].

Deployment of the mass media for disseminating self-help measures and communicating survivor experiences to the general populace has been recommended for the reduction of stress during this pandemic [6]. In the SSA context, these approaches could improve the coping strategies of the populace, particularly those who are susceptible to biological or psychosocial stressors. Additionally, previous studies have shown that many communities in SSA rely on social resources for dealing with mental health issues as the utilization of orthodox mental health care services is generally low [12,20]. Hence, this paper aimed to investigate mental health and emotional effects of COVID-19 pandemic across SSA region as well as to identify those at greater risk, who could be targeted for improved mental and emotional health, during this and future pandemic.

## 2. Methodology

### 2.1. Setting and Study Participants

This online survey was conducted via Survey monkey between 27 April and 17 May 2020, corresponding to the mandatory lockdown period in most SSA countries. At the time, it was not feasible to undertake a conventional Africa-wide community-based sampling survey due to the lockdown and restricted movement. A one-page project information statement that doubled as a recruitment poster was posted/reposted to WhatsApp chat groups and individual WhatsApp accounts. The page also had the contact email of the lead researcher if participants needed further information regarding any part of the study. A link to the online survey was provided. Recipients were further encouraged to send on or ‘snowball’ the e-link of the survey to other WhatsApp groups that they knew as well as to friends and other social media outlets. We also sent out the link by email to selected groups and individuals in all of the target countries relying on the authors’ networks with collaborating academics and local people living in SSA countries. Survey responses were saved and stored on survey monkey regional data center, and anonymized data was retrieved at the end of the study period for analysis.

### 2.2. Survey Design

The survey instrument was adapted and developed from WHO-recommended questions used in a previous survey (Cronbach’s alpha coefficients of 0.74) [21] and had a brief overview of the context, purpose, procedures, nature of participation, privacy and confidentiality statements and notes to be filled out. For this study, we used a twenty-four item self-administered online questionnaire (Appendix A) which was divided into five sections (Sociodemographic and household factors, public attitudes toward compliance to COVID-19, mitigation measures, and risk perception for contracting COVID-19 infection). All questions relating to sociodemographic and household factors were mandatory.

Prior to the launching of the survey, a pilot study was conducted to ensure clarity and understanding as well as to determine the duration for completing the questionnaire. Participants (*n* = 10) who took part in the pilot were not part of the research team and did not participate in the final survey as well. In order to minimize bias, responses to the risk perception and attitude items of the online survey used a five-point Likert scale with provisions for neutral responses, so that the answers were not influenced in one way or another. The participants did not receive any incentives; their responses were voluntary and anonymous. The Kudar-Richardson 20 (KR-20) Cronbach’s alpha coefficient measuring internal consistency reliability for measures with dichotomous responses for the five mental health symptoms ranged from 0.70 to 0.74, indicating satisfactory consistency.

### 2.3. Inclusion and Exclusion Criteria

To be eligible for participation, the participants had to be SSA nationals either living abroad or in their countries of origin, which included Ghana, Cameroon (mostly distributed to the English-speaking individuals), Nigeria, South Africa, Tanzania, Kenya and Uganda, aged 18 years or more, and able to provide online consent. Non-SSA participants were excluded from this study.

### 2.4. Consent and Ethical Consideration

The Human Research Ethics Committee of the Cross-River State Ministry of Health, Nigeria approved this study (number of ethical approval: CRSMOH/RP/REC/2020/116). The study was carried out in accordance with the Helsinki Declaration for Human Research. The section also advised participants not to complete the survey a second time if they had already done so, and that only those aged 18 years were eligible to participate. The confidentiality of participants was assured in that no identifying information was obtained from participants.

This was followed by a consent section where participants were required to voluntarily respond with either a ”yes” or ”no” to the question inquiring whether they voluntarily agreed to participate in the survey. Participants who answered “yes” were directed to complete the survey. All participants gave written informed consent before participation in this study.

### 2.5. Data Analysis

#### 2.5.1. Outcome Variables

The outcome variables in this study were derived from the item asking participants: ”how do you feel about the COVID-19 lockdown measures?“ (item 16 of Appendix A). The measures were, ”frustrated”, ”angry’, ”bored”, ”anxious” and ”worried” about COVID-19. Each of these five outcome variables were coded as binary, ”1” for yes and ”0” for no.

#### 2.5.2. Confounding Variables

The confounding variables used in this study included the demographic characteristics (age group, gender, marital status, and place of current residence, education, employment, occupation, religion; household factors (whether they lived alone and the number of household members which was an open-ended question, see item 12 of Appendix A). The items required a true/false, yes/no response with an additional “I don’t know/unsure” option provided. The public attitude towards COVID-19 mitigation practice variables were obtained from questions on whether the respondents practiced self-isolation, or home quarantine, and adhered to the precautionary public health measures such as, avoiding crowded places or religious events, use of face mask when leaving their homes, and practicing hand hygiene (washing hands with soap for at least 20 s each time or using hand sanitizers). These items were added to identify the effect of compliance to the mitigation practices put in place during the lockdown period to prevent the spread of the virus. For these variables, each question used a Likert scale with five levels with scores for each item ranging from 0 (lowest) to 4 (highest). As with epidemiological studies, the Likert scales were dichotomized to aid epidemiological interpretations and to describe the type of outcome under study (prevalence study and odds ratios). The risk perception variables were derived from questions on whether or not the respondents thought they were “at risk of becoming infected”, “at risk of dying from the infection”, “at risk of becoming severely infected”, “how worried they were because of COVID-19” if they thought “the infection would continue in their country” and how concerned they were of the possibility of being infected. These were included because individuals who perceived the risk as severe are more likely to reduce the spread of the virus.

### 2.6. Statistical Analysis

Initial analyses involved frequency tabulations of all confounding factors in the study population presented in Table 1. This was followed by cross-tabulation to determine the prevalence and their corresponding 95% confidence intervals (CIs) of the mental health symptoms such as feeling anxious and emotional features that included being bored, frustrated, worried and angry. Univariate logistic regression was performed to examine the independent association between the five mental and emotional health symptoms (feeling bored, anxious, frustrated, worried and angry) and confounding factors (see Table 1 for details).

Multivariable logistic regression was also carried out to determine factors associated with the five mental and emotional health symptoms. The odds ratios (OR) with their 95% CIs were calculated to assess the adjusted odds of the confounding variables and those with *p*-value < 0.05 were considered as factors associated with the five variables (see, bolded adjusted OR and their 95% CIs in Table 2). All analyses were conducted using STATA/MP version 14.1 (Stata Corp 2015, College Station, TX, USA).

## 3. Results

### 3.1. Sample Characteristics

Table 1 presents the details of the demographic variables of participants in this study. A total of 2005 adults from SSA completed the survey, about half of them were males, not married, and many were aged 18 to 38 years. At the time of this study, majority of the respondents lived in their SSA countries of origin, particularly in West Africa, were non-healthcare workers, had completed at least a bachelor’s degree, were employed and lived alone. Due to the web-based design, it was not possible to estimate how many persons were reached by social network advertisement and no response rate could be estimated.

More than one third experienced self-quarantine due to COVID-19 and nearly all respondents (94.3%) were concerned about contracting COVID-19 while some (19.4%) thought they were at high risk of dying from the infection. A high percentage of respondents believed that COVID-19 would not continue after the lockdown (1167, 63.9%). Further details are presented in Table 1.

### 3.2. Prevalence of Mental Health/Emotional Symptoms

The prevalence of self-reported mental health and emotional issues and their 95% CIs are shown in Figure 1. The prevalence was highest for respondents who were bored (57.5%, 95% CI 55.2%, 59.7%]), followed by those who felt anxious (59.1%, 95% CI 56.7%, 61.5%) and worried (57.5%, 95% CI 55.2%, 59.7%) about the pandemic. Overall, more than 52.2% of the participants reported mental and emotional health symptoms.

### 3.3. Univariate Analysis of Factors Associated with Mental Health Symptoms

The univariate analysis of factors associated with the symptoms of mental health and emotional effects in the study population is presented in Table 2. Living in Central Africa, with six or more people in the household was associated with increased odds of feeling bored, frustrated, angry and anxious among the respondents. Those who lived with more than six persons in the household showed significantly higher odds for all the dependent variables except for "feeling worried" and East African respondents had remarkably higher odds of feeling frustrated due to COVID-19. Compared to men, women were more likely to feel bored (OR 1.28, 95% CI 1.02, 1.59) and anxious (OR 1.24, 95% CI 1.02, 1.53) and those who were not married were more likely to feel frustrated (OR 1.25, 95% CI 1.02, 1.52) and angry (OR 1.30, 95CI 1.02, 1.66) compared to the married respondents. Higher odds of feeling ‘angry’ was found among those who were unemployed (OR 1.29, 95% CI 1.04, 1.59), and among respondents who thought that COVID-19 would not continue in their respective countries after the lockdown (OR 1.40, 95% CI 1.08, 1.81).

Individuals who were concerned that they or their family members could be infected with COVID-19 were less likely to feel worried and anxious about contracting the infection. Similarly, participants who felt at lower risk of being infected (OR 0.34, 95% CI 0.27, 0.41), or being severely infected (OR 0.26, 95% CI 0.20, 0.33) and those who thought their risk of dying from the infection was low (OR 0.18, 95% CI 0.14, 0.25), were less likely to worry about COVID-19, in this study.

### 3.4. Multivariate Analysis of Factors Associated with Mental Health/Emotional Symptoms

Table 3 shows the factors associated with the symptoms of mental and emotional health, after adjusting for all potential covariates. Age became a significant factor influencing the respondents’ experience of mental health symptoms. Participants aged 29–38 years had higher odds for feeling bored (aOR 1.81, 95% CI 1.05, 3.10), and frustrated (aOR 1.95, 95% CI 1.20, 3.19), while those aged 39-48yrs (aOR 2.09, 95% CI 1.22, 3.56) were more likely to feel frustrated due to COVID-19 compared to younger participants (18–28years). Central African respondents reported higher odds of feeling frustrated (aOR 1.49, 95% CI 1.01, 2.19), angry (aOR 2.12, 95% CI 1.37, 3.29), and anxious (aOR 1.60, 95% CI 1.05, 2.43), whereas respondents from Southern African countries reported higher odds of feeling frustrated (aOR 1.46, 95% CI 1.06, 2.00), compared to those from the West African countries. Other factors associated with higher odds of mental and emotional health symptoms in this study included being unmarried, being unemployed, living with six or more people in the household during the pandemic, perception of low risk of contracting the infection and the thought that COVID-19 will not continue after the lockdown.

Overall, respondents who perceived a low risk of being infected by COVID-19 were less likely to be worried about the disease and those from the Southern African countries were less likely to feel bored during the pandemic (aOR 0.59, 95% CI 0.42, 0.82), compared to the respondents from West Africa.

## 4. Discussion

The current study explored both mental and emotional health symptoms among SSA respondents, during the COVID-19 lockdown. This is the first study using a web-based cross-sectional survey to examine the prevalence and factors associated with mental and emotional health symptoms of COVID-19 in SSA. This study found that the COVID-19 pandemic had a major impact on the mental and emotional health of respondents in SSAs (including health care workers). More than half of the respondents reported feeling anxious, worried, frustrated and bored, whereas approximately one in four respondents reported feeling angry during the COVID-19 lockdown period. The study also revealed that those older than 28 years who lived in the Central and Southern African countries, respondents who were not married, the unemployed, as well as those who lived with more than six persons in a household, had higher odds of mental and emotional health symptoms during the COVID-19 lockdown period. In addition, respondents who felt that at lower risk of being infected by the virus and those who did not think that COVID-19 will continue after the lockdown, were more likely to feel angry about the pandemic. The study also found that people working in health care sectors were less likely to report mental and emotional health symptoms during the COVID-19 pandemic.

Compared with previous rates of any mental health/emotional symptoms in Low and Middle-Income Countries (LMIC) and SSA countries (10–20% at any one time [20,22]), the impact of the pandemic on the state of mental health (which includes emotional and psychological well-being) of SSAs in this study was profound and increased by three folds. Reports from China [23,24], the USA (53% reported feeling anxious and stress relating to the coronavirus) [25], India (30.5% reported depression) [26] and Italy (41.6% reported moderate stress) [27], found much lower prevalence of mental illness/emotional symptoms than the present study. Although COVID-19 infections and deaths are lower in SSA region than other regions, the higher prevalence of mental health symptoms found in this study suggests that the SSA population could be particularly vulnerable to emotional distress in the current pandemic. Given the already existing situations of poverty, unemployment and weak health systems [20] and the unavailability of effective treatment, it is expected that the rate of mental and emotional health symptoms will increase. This suggestion is substantiated in Table 2, whereby it is observed that respondents were more worried about the likelihood of COVID-19 continuing (63.9%) than their risk of becoming severely infected (25.8%) or dying (19.4%) from the disease. This finding is consistent with a recent study from the UK, which found that the citizens were more concerned about how societal changes would influence their psychological and financial wellbeing, than their risk of becoming unwell with the virus [28]. Prevention efforts such as screening for mental health and emotional problems and psychoeducation [29] focusing on the identified groups at risk for adverse psychosocial outcomes are needed.

The higher odds of mental illness/emotional symptoms among health care workers (HCWs) than non-healthcare workers (NHCWs) found in previous studies among Chinese residents [9,11] were not found in this study. Rather, we found that HCWs were less likely to report any COVID-19 related mental health symptom than NHCWs, particularly with respect to feeling ‘angry’. Despite the lower odds of mental health/emotional symptoms among HCWs, they are particularly vulnerable to emotional distress in the current pandemic. This is due to their level of exposure to the virus, concern about being infected and caring for their loved ones, shortages of personal protective equipment (PPE), longer work hours, and involvement in emotionally and ethically fraught resource-allocation decisions [30]. HCWs should be monitored for a change in routine and behavior. Similar to the previous reports from Italy [27] and China [9,31], the present study found a significantly higher odds of mental health symptoms among women than men during the pandemic (Table 3). This finding may be explained in part by the report that women including those who are pregnant or have young children were more likely to develop the fear of becoming infected or transmitting the virus [32].

As part of the measures to deal with mental health and psychological issues during the COVID-19 pandemic, the WHO had encouraged individuals to stay with friends and families [10]. This guideline was supported by our findings of higher odds for feeling bored, frustrated, and angry among respondents who were single. On the other hand, living with more than six persons in the household was associated with higher odds of mental and emotional health symptoms of feeling anxious, angry, and frustrated. This may also suggest that living with many people in the household could make people feel more anxious. Evidence from a systematic review and meta-analysis on household transmission studies of COVID-19 found that the risk of infection is 10 times higher among household contacts than other contacts [33]. These trends have challenged the traditional SSA social structure of communalism, a crucial socio-cultural factor, whose maintenance could be important in dealing with mental health issues in the SSA context [6]. Many communities in SSA rely on social resources for dealing with mental health issues as utilization of orthodox mental health care services is generally low [12,20]. Some of the resources people access for relief from mental problems within the SSA context include keeping in touch with others, attending faith and religious events, and engaging in prayers [6].

In the wake of the COVID-19 lockdowns in SSA, access to social resources was limited, and alternative ways of delivering mental health resources were needed [12]. Prior to COVID-19, there was already a huge gap of unmet mental health services for older adults in SSA [12,34] which is fueled by factors such as stigma, poor awareness that older adults suffer from mental illness, deficient primary health care services, inadequate community healthcare workers and psychogeriatricians [35]. In addition to these factors, the low levels of digital literacy in SSA hindered the deployment of online or virtual mental health service delivery [6] as alternatives to overcome disruptions to in-person services. While more than 80% of high-income countries have started the utilization of alternative mental health interventions to bridge gaps in mental health services, the patronage by low-income countries has been less than 50% [7]. In line with WHO’s guideline, SSA countries must allocate resources to mental health as an integral component of their response and recovery strategies. Utilization of the mass media to share survivor experiences to mental health patients and the public could be a good alternative for delivering counselling measures [6]. Additionally, educational campaigns are needed to increase the awareness and enlightenment of the public regarding the fact that older adults can suffer from mental illnesses and recognize the benefits of orthodox management of mental illnesses [12].

The present findings indicate that individuals who perceived themselves or their family members to be at a lower risk of being infected by COVID-19 or dying from the disease reported lower odds for worrying about COVID-19. Strong coping mechanisms are necessary to deal with mental health and emotional issues during a pandemic, and one of these coping strategies is to be less concerned about the consequences and impact of the disease and remain cautious. According to the WHO, people should reduce the amount of information they receive about COVID-19 to reduce feeling anxious [10]. Consistent with our finding, a study in China found that people who spent too much time thinking about disease outbreak were more likely to develop symptoms of anxiety [11]. This may explain the higher prevalence of mental health/emotional symptoms, which we found among the unemployed respondents. The mental health and psychological impact of the pandemic because of rising economic recession and massive job losses within the context of struggling economies cannot be underestimated. Therefore, mental health support services should be an integral part of the disease response strategy in SSA.

## 5. Limitations and Strengths

This study has some limitations. First, the data was collected using an online survey and may not be a true reflection of the opinion of other SSAs living in rural areas where internet penetration and connectivity remain relatively low [36] or the older people who are less likely to use the internet. However, the increase in the use of internet recorded among the general population during the pandemic meant that many people may have participated [37]. Also, this was the only reliable, cost effective means to disseminate information at the time of this study and obtain real-time data on the current situation. Second, the survey was available only in English, making it difficult for people living in French-speaking countries including some part of Cameroon, to participate. Third, there was limited participation of East African respondents in this study, which may be attributed to the government instructions for residents to refrain from giving out information regarding the pandemic. Fourth, the research methodology did not allow us to reach people with medically examined mental health symptoms; therefore, the provision of the results may not fully reflect the severity of the mental health symptoms among SSA population. Fifth, this study did not use the tools designed specifically for the COVID-19 pandemic, such as the coronavirus anxiety scale (CAS). Future studies using the tools developed especially for the COVID-19 pandemic will provide a concrete finding and facilitate the demand for a focused public health initiative. Another limitation peculiar to web-based surveys was the inability to verify the eligibility of the participants and the validity of their responses. Despite these limitations, this is the first study to provide comprehensive evidence of the mental health impact of the pandemic across SSA region. With a web-based questionnaire, the study was able to assess the prevalence of mental health symptoms among SSA respondents, while maintaining the WHO recommended “social distance” during the COVID-19 pandemic, which otherwise would be impossible. Furthermore, using a robust analysis, we were able to minimize bias by controlling for all potential confounders in the analysis. That no incentives were not given to participant’s ensured that their participation and response rates were not influenced [38].

## 6. Conclusions

In conclusion, amidst relatively lower disease and death ratios, this study highlights a high prevalence of mental health and emotional symptoms during COVID-19 in Sub Sahara African region. Despite the lack of a baseline mental health study of the study population prior to COVID-19, the findings strongly suggest markedly elevated mental health symptoms whose rates are consistent with those of other study populations worldwide [39]. Such a high impact of the disease may be related to the weak health systems and low access to alternative mental health service delivery within the sub-region. While three out of every four persons surveyed reported feeling bored, about one in every two persons, felt frustrated and worried about the lockdown. Southern and Central Africans had a greater risk of mental health and emotional symptoms during COVID-19 pandemic. Implementing community-based strategies to support resilience among these psychologically vulnerable individuals such as the older adults, those who neither are married nor employed, as well as people living with many household members during the COVID-19 crisis is fundamental for the SSA communities. The psychological impact of fear and feeling anxious induced by the rapid spread of pandemic needs to be clearly recognized as a public health priority for both authorities and policymakers who should rapidly adopt clear behavioral strategies to reduce the burden of disease, plan for long-term fallout of the disease and the dramatic mental health consequences of this outbreak. Most importantly, mental health service resources must be as an integral component of SSA governments’ response and recovery strategies of the COVID-19 pandemic.

## Figures and Tables

**Figure 1 ijerph-18-00899-f001:**
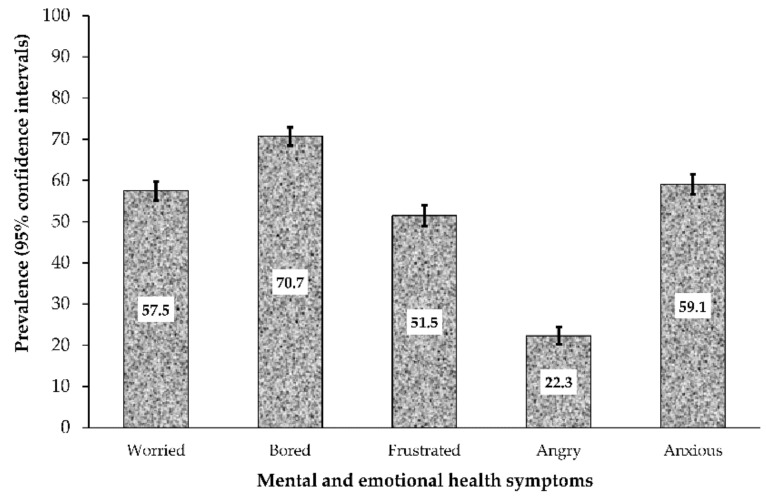
Prevalence of mental health and emotional effects in Sub Sahara African respondents (*n* = 2005) during the COVID-19 Pandemic. Error bars are 95% confidence interval.

**Table 1 ijerph-18-00899-t001:** Characteristics of the study population (*N* = 2005).

Variables	Number	Percentages
Sociodemographic characteristics		
Place of Origin (*n* = 1969)		
West Africa	1108	56.27
East Africa	209	10.61
Central Africa	251	12.75
Southern Africa	401	20.37
Place of residence		
Africa	1855	92.52
Diaspora	150	7.48
Age in years (*n* = 1988)		
18–28 years	775	38.98
29–38	530	26.66
39–48	441	22.18
49+ years	242	12.17
Sex (*n* = 1991)		
Men	1099	55.20
Women	892	44.80
Marital Status (*n* = 1995)		
Married	879	44.06
Not married ^†^	1116	55.94
Highest level of Education (*n* = 1997)		
Postgraduate Degree (Masters/PhD)	642	32.15
Bachelor’s degree	1090	54.58
Secondary/Primary	265	13.27
Employment status (*n* = 2000)		
Employed	1321	66.05
Unemployed	679	33.95
Occupation type (*n* = 1904)		
Non-healthcare	1471	77.26
Healthcare	433	22.74
Religion (*n* = 1995)	1995	
Christianity	1763	88.37
Islam/others †	232	11.63
Household factors		
Do you live alone during COVID-19 (*n* = 1996)		
No	1624	81.36
Yes	372	18.64
Number living together (*n* = 1775)		
1–3 people	506	28.83
4–6 people	908	51.74
6+ people	341	19.43
Public attitudes toward compliance with COVID-19		
Practiced self-isolation (*n* = 1801)		
No	1237	68.68
Yes	564	31.32
Home quarantined due to COVID-19 (*n* = 1798)		
No	1091	60.68
Yes	707	39.32
Gone to a crowded event (*n* = 1797)		
No	1550	86.25
Yes	247	13.75
Perception of risk		
Risk of becoming infected (*n* = 1821)		
High	674	37.01
Not high	1147	62.99
Risk of becoming severely infected (*n* = 1823)		
High	471	25.84
Not high	1352	74.16
Risk of dying from infection (*n* = 1818)		
High	352	19.36
Not high	1466	80.64
Possibility of you/family member being affected (*n* = 1794)		
Concerned	1692	94.31
Not Concerned	102	5.69
Likelihood of COVID-19 continuing (*n* = 1827)		
Likely	1167	63.88
Not Likely	660	36.12

Note: Total count 2005 unless otherwise given in brackets. ^†^ = single, previously married, divorced or widowed.

**Table 2 ijerph-18-00899-t002:** Univariate analysis of factors associated with five mental health/emotional symptoms of COVID-19 among Sub-Sahara Africans during the lockdown.

Variables	Worried		Bored		Frustrated		Angry		Anxious	
	OR	95% CI	OR	95% CI	OR	95% CI	OR	95% CI	OR	95% CI
Demography										
Place of Origin										
West Africa	1.00		1.00		1.00		1.00		1.00	-
East Africa	1.34	[0.97, 1.85]	0.61	[0.43, 0.86]	**1.56**	**[1.12, 2.18]**	1.27	[0.85, 1.89]	1.09	[0.78, 1.52]
Central Africa	1.19	[0.89, 1.60]	**1.56**	**[1.06, 2.30]**	**1.64**	**[1.20, 2.24]**	**2.38**	**[1.69, 3.35]**	**1.98**	**[1.41, 2.79]**
Southern Africa	1.21	[0.95, 1.54]	0.60	[0.4, 0.79]	**1.48**	**[1.14, 1.92]**	1.05	[0.76, 1.46	0.99	[0.76, 1.28]
Place of residence										
Africa	1.00		1.00		1.00		1.00		1.00	
Diaspora	0.76	[0.53, 1.08]	1.23	[0.80, 1.89]	0.82	[0.56, 1.19]	0.90	[0.56, 1.44]	0.84	[0.57, 1.24]
Age in years		-	-	-	-	-	-	-	-	-
18–28	1.00		1.00		1.00		1.00		1.00	
29–38	1.17	[0.91, 1.51]	1.12	[0.83, 1.50]	0.86	[0.66, 1.13]	0.85	[0.61, 1.18]	0.92	[0.70, 1.21]
39–48	1.09	[0.81, 1.48]	1.03	[0.72, 1.48]	1.10	[0.80, 1.53]	0.91	[0.61, 1.35]	0.75	[0.54, 1.04]
49+	1.04	[0.83, 1.32]	0.13	[0.71, 1.21]	0.84	[0.66, 1.08]	0.95	[0.71, 1.29]	0.96	[0.75, 1.24]
Sex										
Men	1.00	-	1.00	-	1.00	-	1.00	-	1.00	-
Women	0.98	[0.82, 1.19]	**1.28**	**[1.02, 1.59]**	0.98	[0.80, 1.20]	0.89	[0.70, 1.14]	**1.24**	**[1.02, 1.53]**
Marital Status	-	-	-	-	-	-	-	-	-	-
Married	1.00	-	1.00	-	1.00	-	1.00	-	1.00	-
Not married^†^	1.06	[0.88, 1.28]	1.12	[0.90, 1.40]	**1.25**	**[1.02, 1.52]**	**1.30**	**[1.02, 1.66]**	1.16	[0.95, 1.42]
Highest level of Education	-	-	-	-	-	-	-	-	-	-
Postgraduate Degree (Masters /PhD)	1.00		1.00	-	1.00	-	1.00	-	1.00	-
Bachelor’s degree	**1.24**	**[1.01, 1.52]**	**0.72**	**[0.56, 0.91]**	1.12	[0.90, 1.40]	0.94	[0.72, 1.23]	1.03	[0.82, 1.29]
Secondary/Primary	1.02	[0.75, 1.39]	1.04	[0.72, 1.51]	1.23	[0.89, 1.69]	0.74	[0.49, 1.12]	0.98	[0.71, 1.36]
Employment status	-	-	-	-	-	-	-	-	-	-
Employed	1.00	-	1.00	-	1.00	-	1.00	-	1.00	-
Unemployed	1.09	[0.90, 1.33]	1.02	[0.81, 1.28]	**1.29**	**[1.04, 1.59]**	1.01	[0.78, 1.30]	1.01	[0.81, 1.24]
Occupation type	-	-	-	-	-	-	-	-	-	-
Non-healthcare	1.00	-	1.00	-	1.00	-	1.00	-	1.00	-
Healthcare	1.10	[0.88, 1.39]	0.90	[0.70, 1.17]	**0.78**	**[0.61, 0.99]**	**0.64**	**[0.46, 0.88]**	1.17	[0.91, 1.49]
Religion	-	-	-	-	-	-	-	-	-	-
Christianity	1.00	-	1.00	-	-	-	-	-	-	-
Islam/others†	1.20	[0.89, 1.61]	0.91	[0.65, 1.28]	0.89	[0.65, 1.22]	1.02	[0.70, 1.49]	0.89	[0.65, 1.22]
Household factors										
Number living together		-	-	-	-	-	-	-	-	-
1–3 people	1.00	-	1.00		1.00		1.00		1.00	
4–6 people	1.13	[0.90, 1.43]	1.15	[0.89, 1.49]	1.07	[0.83, 1.36]	1.27	[0.93, 1.72]	1.14	[0.89, 1.46]
6+ people	0.97	[0.73, 1.29]	**1.57**	**[1.11, 2.23]**	**1.42**	**[1.04, 1.95]**	**1.64**	**[1.12, 2.37]**	**1.39**	**[1.01, 1.93]**
Attended crowded event										
No	1.00	-	1.00	-	1.00	-	1.00	-	1.00	-
Yes	1.15	[0.87, 1.51]	1.02	[0.73, 1.42]	1.06	[0.78, 1.42]	0.99	[0.68, 1.43]	1.20	[0.87, 1.68]
Public attitudes towards compliance to COVID-19										
Self-Isolation										
No	1.00	-	1.00	-	1.00	-	1.00	-	1.00	-
Yes	1.07	[0.87, 1.31]	0.96	[0.75, 1.22]	0.97	[0.77, 1.21]	0.80	[0.60, 1.06]	0.85	[0.67, 1.06]
Home quarantined due to COVID-19	-	-	-	-	-	-	-	-	-	-
No	1.00	-	1.00	-	1.00	-	1.00	-	1.00	-
Yes	1.03	[0.85, 1.25]	1.04	[0.82, 1.31]	1.01	[0.81, 1.25]	0.86	[0.66, 1.11]	0.96	[0.77, 1.20]
Perception of risk										
Risk of becoming infected	-	-	-	-	-	-	-	-	-	-
High	1.00	-	1.00	-	1.00	-	1.00	-	1.00	-
Not high	**0.34**	**[0.27, 0.41]**	**1.16**	[0.92, 1.46]	1.22	[0.99, 1.52]	**1.51**	**[1.15, 1.99]**	1.12	[0.90, 1.39]
Risk of becoming severely infected	-	-	-	-	-	-	-	-	-	-
High	1.00	-	1.00	-	1.00	-	1.00	-	1.00	-
Not high	**0.26**	**[0.20, 0.33]**	1.11	[0.86, 1.43]	1.05	[0.83, 1.33]	1.14	[0.85, 1.53]	1.06	[0.84, 1.35]
Risk of dying from infection	-	-	-	-	-	-	-	-	-	-
High	1.00	-	1.00	-	1.00	-	1.00	-	1.00	-
Not high	**0.18**	**[0.14, 0.25]**	1.30	[0.98, 1.72]	1.12	[0.86, 1.46]	1.06	[0.77, 1.47]	1.08	[0.83, 1.41]
Possibility of you/family member being affected	-	-	-	-	-	-	-	-	-	-
Concerned	1.00	-	1.00	-	1.00	-	1.00	-	1.00	-
Not Concerned	**0.17**	**[0.10, 0.27]**	0.92	[0.58, 1.48]	0.80	[0.51, 1.24]	0.70	[0.39, 1.26]	**0.63**	**[0.40, 0.98]**
Likelihood of COVID-19 continuing	-	-	-	-	-	-	-	-	-	-
Likely	1.00	-	1.00	-	1.00	-	1.00	-	1.00	-
Not Likely	**0.62**	**[0.51,0.75]**	1.12	[0.58, 1.48]	1.17	[0.94, 1.45]	**1.40**	**[1.08, 1.81]**	1.08	[0.86, 1.34]

OR = odds ratio; Bolded: confidence intervals (CIs) are significant. ^†^ = single, previously married, divorced or widowed.

**Table 3 ijerph-18-00899-t003:** Multivariate analysis of factors associated with mental health/emotional impact of COVID-19 among Sub-Sahara Africans during the lockdown.

Variables	Worried		Bored		Frustrated		Angry		Anxious	
	aOR	95% CI	aOR	95% CI	aOR	95% CI	aOR	95% CI	aOR	95% CI
Demography										
Place of Origin										
West Africa	1.00	-	1.00	-	1.00	-	1.00	-	1.00	-
East Africa	1.13	[0.77, 1.66]	**0.48**	**[0.32, 0.72]**	1.16	[0.78, 171]	1.05	[0.65, 1.70]	0.99	[0.67, 1.47]
Central Africa	1.07	[0.74, 1.56]	1.37	[0.85, 2.21]	**1.49**	**[1.01, 2.19]**	**2.12**	**[1.37, 3.29]**	**1.60**	**[1.05, 2.43]**
Southern Africa	1.17	[0.87, 1.58]	0.59	**[0.42, 0.82]**	**1.46**	**[1.06, 2.00]**	0.87	[0.58, 1.31]	0.90	[0.65, 1.23]
Place of residence										
Africa	1.00	-	1.00	-	1.00	-	1.00	-	1.00	-
Diaspora	1.00	[0.59, 1.69]	1.70	[0.86, 3.38]	0.94	[0.53, 1.64]	0.86	[0.42, 1.78]	0.88	[0.49, 1.58]
Age in years										
18–28	1.00	-	1.00	-	1.00	-	1.00	-	1.00	-
29–38	1.12	[0.76, 1.65]	1.27	[0.83, 1.96]	**1.56**	**[1.03, 2.34]**	1.11	[0.69, 1.79	0.93	[0.62, 1.40]
39–48	1.49	[0.94, 2.35]	**1.81**	**[1.05, 3.10]**	**1.95**	**[1.20, 3.19]**	1.01	[0.56, 1.81]	0.93	[0.57,1.52]
49+	1.19	[0.72, 1.98]	1.58	[0.87, 2.81]	**2.09**	**[1.22, 3.56]**	0.95	[0.49, 1.82]	0.69	[0.40, 1.17]
Sex										
Men	1.00	-	1.00	-	1.00	-	1.00	-	1.00	-
Women	1.05	[0.82, 1.33]	1.15	[0.88, 1.52]	0.78	[0.60,1.00]	**0.68**	**[0.50, 0.93]**	1.17	[0.90, 1.51]
Marital Status	-	-	-	-	-	-	-	-	-	-
Married	1.00	-	1.00	-	1.00	-	1.00	-	1.00	-
Not married ^†^	1.10	[0.79, 1.52]	**1.53**	**[1.04, 2.42]**	**1.65**	**[1.16, 2.33]**	**1.81**	**[1.19, 2.75]**	1.40	[0.99, 1.99]
Highest level of Education										
Postgraduate Degree (Masters/PhD)	1.00	-	1.00	-	1.00	-	1.00	-	1.00	-
Bachelor’s degree	1.29	[0.96, 1.74]	0.79	[0.56, 1.11]	1.19	[0.87, 1.63]	1.01	[0.69, 1.49]	0.94	[0.69, 1.28]
Secondary/Primary	1.12	[0.70, 1.80]	1.24	[0.72, 2.14]	1.01	[0.62, 1.64]	0.55	[0.29, 1.04]	0.64	[0.40, 1.05]
Employment status										
Employed	1.00	-	1.00	-	1.00	-	1.00	-	1.00	-
Unemployed	**1.44**	**[1.03, 2.02]**	0.94	[0.64, 1.37]	**1.45**	**[1.01, 2.07]**	0.81	[0.53, 1.24]	0.78	[0.54, 1.13]
Occupation type										
Non-healthcare	1.00	-	1.00	-	1.00	-	1.00	-	1.00	-
Healthcare	1.02	[0.77, 1.36]	0.98	[0.71, 1.34]	0.75	[0.55, 1.00]	**0.60**	**[0.41, 0.90]**	1.18	[0.87,1.59]
Religion										
Christianity	1.00	-	1.00	-	-	-	-	-	-	-
Islam/others†	1.15	[0.78, 1.69]	0.93	[0.60, 1.45]	0.99	[0.66, 1.47]	1.26	[0.77, 2.05]	0.89	[0.59, 1.34]
Household factors										
Number living together										
<3 people	1.00		1.00	-	1.00	-	1.00	-	1.00	-
4–6 people	1.12	[0.85,1.47]	1.27	[0.94, 1.72]	1.03	[0.78, 1.38]	1.17	[0.81, 1.66]	1.23	[0.92, 1.63]
6+ people	0.94	[0.66, 1.32]	**1.70**	**[1.13, 2.56]**	1.42	[0.99, 2.05]	1.20	[0.77, 1.87]	1.31	[0.90, 1.90]
Attended crowded event										
No	1.00	-	1.00	-	1.00	-	1.00	-	1.00	-
Yes	1.09	[0.76, 1.56]	1.02	[0.68, 1.56]	1.11	[0.76, 1.60]	0.88	[0.55, 1.40]	1.06	[0.72, 1.57]
Public attitudes toward compliance to COVID-19										
Self-Isolation										
No	1.00	-	1.00		1.00		1.00		1.00	
Yes	0.96	[0.72, 1.28]	0.86	[0.62, 1.18]	0.86	[0.64, 1.15]	0.69	[0.48, 1.00]	0.85	[0.63, 1.14]
Home quarantined due to COVID-19										
No	1.00	-	1.00	-	1.00	-	1.00	-	1.00	-
Yes	1.01	[0.77, 1.31]	0.95	[0.70, 1.29]	1.05	[0.79, 1.38]	1.16	[0.83, 1.63]	0.97	[0.73, 1.29]
Perception of risk										
Risk of becoming infected										
High	1.00	-	1.00	-	1.00	-	1.00	-	1.00	-
Not high	**0.57**	**[0.45, 0.80]**	0.94	[0.64, 1.39]	1.32	[0.93, 1.88]	**1.84**	**[1.16, 2.93]**	1.11	[0.78, 1.59]
Risk of becoming severely infected										
High	1.00	-	1.00		1.00		1.00		1.00	
Not high	**0.60**	**[0.39, 0.93]**	0.91	[0.57, 1.48]	0.84	[0.54, 1.88]	0.84	[0.47, 1.48]	0.98	[0.63, 1.52]
Risk of dying from infection										
High	1.00	-	1.00	-	1.00	-	1.00	-	1.00	-
Not high	**0.38**	**[0.25, 0.58]**	1.49	[0.97, 2.30]	1.18	[0.79, 1.76]	0.89	[0.53, 1.48]	1.19	[0.80, 1.78]
Possibility of you/family member being affected										
Concerned	1.00	-	1.00	-	1.00	-	1.00	-	1.00	-
Not Concerned	**0.17**	**[0.09, 0.33]**	1.06	[0.57, 1.97]	0.82	[0.46, 1.45]	0.89	[0.45, 1.79]	0.64	[0.36, 1.14]
Likelihood of COVID-19 continuing										
Likely	1.00	-	1.00	-	1.00	-	1.00	-	1.00	-
Not Likely	**0.76**	**[0.59, 0.97]**	1.08	[0.81, 1.45]	1.27	[0.98, 1.66]	**1.49**	**[1.08, 2.04]**	1.17	[0.89, 1.53]

aOR = adjusted odds ratio; Bolded: confidence intervals (CIs) are significant. ^†^ = single, previously married, divorced or widowed.

## Data Availability

The data presented in this study are available on request from the corresponding author.

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
