# Peer review of "Prevalence and Factors Associated with Mental and Emotional Health Outcomes among Africans during the COVID-19 Lockdown Period—A Web-based Cross-Sectional Study"

_ijerph, 2021, doi:10.3390/ijerph18030899_

Round 1
Reviewer 1 Report
Review of “Prevalence and factors associated with mental health outcomes among Africans during the COVID-19 lockdown period”
The main problem with this paper is that these authors are trying to equate emotions such as boredom, frustration, and anger with mental health variables. These variables are clearly not in the same league as their other two variables of anxiety and worry. If they want to keep all of the variables, then simply change the title and the description statements in the text and refer to the variables as responses, attitudes, emotions, or reactions, as these are clearly not mental health variables. For example, the second paragraph of the Discussion section describes mental health variables of stress and depression; boredom and frustration are not in the same category.
Additional points:
- line 41, reference is made to diaspora but are not discussed in the method section. I would suggest removing
- line 48, “any mental” should be “any of the mental”
- line 52, insert “a” before “low risk”
- lines 100-101, state that the values were not meeting the prediction at the time of writing
- line 125, insert “of” before “the target”
- line 130, insert “a” before “previous”
- line 138, delete “shown” after “questionnaire”
- lines 141-142, what scales are the authors referring to? They should be specific which items were being combined for the alpha values
- section 2.3.1, specify that they are referring to item 16 in Table 1
- line 268, specify that number of household members was an open-ended question (or at least appears to be open-ended in question 12 in Table 1)
- line 282, insert “as real” or “as severe” after “risk” (the sentence is incomplete)
- section 3.1, you do not need to repeat values in the text which appear in the table
- line 312, change “will” to “would”
Author Response
Response to Reviewer 1 comments and Suggestions
Dear Editor-in-chief,
Thank you for allowing us to revise and re-submit our manuscript. We are incredibly grateful for your insightful feedback that has strengthened our manuscript.
We have addressed all the issues raised by the reviewers in the revised version of the manuscript. Where we have a different opinion from the reviewers, we have clarified our position. All changes made to the revised manuscript have been highlighted in red. Below, we provide a point by point response to your comments and where possible, linked our responses to the appropriate sections in the revised manuscript. We hope that the revisions made have properly addressed your concerns, and that our revised manuscript is acceptable for publication in the journal.
Review of “Prevalence and factors associated with mental health outcomes among Africans during the COVID-19 lockdown period”
|
Review Comments |
Response |
Changes made |
|
Reviewer 1: Main comments |
||
|
The main problem with this paper is that these authors are trying to equate emotions such as boredom, frustration, and anger with mental health variables. These variables are clearly not in the same league as their other two variables of anxiety and worry. If they want to keep all of the variables, then simply change the title and the description statements in the text and refer to the variables as responses, attitudes, emotions, or reactions, as these are clearly not mental health variables. For example, the second paragraph of the Discussion section describes mental health variables of stress and depression; boredom and frustration are not in the same category. |
The reviewer may be referring to mental illness or disorders which are different from mental health and emotional symptoms. This study focused on mental and emotional health symptoms as the mental health symptoms, naturally, include emotional and psychological features.
However, we have modified the title to reflect this and the study type. We also made major changes in the abstract and throughout the manuscript to reflect these changes. |
|
|
Minor comments |
|
|
|
- line 41, reference is made to diaspora but are not discussed in the method section. I would suggest removing |
This has been removed |
|
|
- line 48, “any mental” should be “any of the mental” |
done |
|
|
- line 52, insert “a” before “low risk” |
done |
|
|
- lines 100-101, state that the values were not meeting the prediction at the time of writing |
This has been added |
Line 116 |
|
- line 125, insert “of” before “the target” |
done |
|
|
line 130, insert “a” before “previous” |
done |
|
|
line 138, delete “shown” after “questionnaire” |
done |
|
|
lines 141-142, what scales are the authors referring to? They should be specific which items were being combined for the alpha values |
This has been revised. The section now reads: In order to minimize bias, responses to the risk perception and attitude items of the online survey used a five-point Likert scale with provisions for neutral responses, so that the answers were not influenced in one way or another. The participants did not receive any incentives; their responses were voluntary and anonymous. The Kudar-Richardson 20 (KR-20) Cronbach’s alpha coefficient measuring internal consistency reliability for measures with dichotomous for the five mental health symptoms ranged from 0.70 to 0.74 indicating satisfactory consistency. |
Lines 163-164 |
|
section 2.3.1, specify that they are referring to item 16 in Table 1 |
Done. |
Line 179 |
|
line 268, specify that number of household members was an open-ended question (or at least appears to be open-ended in question 12 in Table 1) |
done |
Line 188 |
|
line 282, insert “as real” or “as severe” after “risk” (the sentence is incomplete) |
done |
Line 203 |
|
section 3.1, you do not need to repeat values in the text which appear in the table |
The values in the text which appear in the table have been removed where necessary |
|
|
line 312, change “will” to “would” |
done |
Line 325 |
Reviewer 2 Report
The study is important, interesting and pioneering, but it requires a reorganization of the sections, in addition to highlighting and deepening even more the arguments related to the objective and results of the study. It is important to follow the style or structure of epidemiological studies.
Title. Write a more attractive one, perhaps written as a question, and indicate the type of study (considering the variables).
Abstract. For clarity, place the percentages next to each symptom or factor. Write the sentence better “mental health symptoms”, because they do not indicate mental health, (this is repeated a few times in the manuscript, so it must also be modified).
Introduction. It must be rewritten. Indicate how similar situations have been addressed in the past and in other countries in the face of the current situation, how the situation has been evaluated in other studies, informing the recommendations of said studies. In addition, it is important to define the study variables (they could be noted in a section on the study instrument). It is important to further develop the argument presented in the penultimate paragraph and the situation of COVID-19 in the African context (paragraph located from line 100 to 106). Write the objective better, in relation to the title and the methodology (considering the variables), which should be developed in the introduction clearly and directly, that is, the relationship between the independent variables and their influence on the dependent variables.
Methodology.
Study environment and participants. Where did you send the questionnaire when you filled it out? Were the answers saved? (This should be indicated in the next section) indicate the characteristics of the participants, as well as the inclusion and exclusion criteria.
Survey design. Place the survey in an appendix and cite it within the text. Explain in more detail how the study was explained to the participants and the aspects of informed consent. If they did not understand something, did they have any contact to find an explanation? How were they indicated about anonymity and how was confidentiality assured? (since they are not indicated in the survey) How did you control that they were over 18 years of age? In addition, does the study comply with any agreement on an ethical standard? Some analysis was carried out on the questionnaire, in addition to extracting Cronbach's alpha, to evaluate its suitability? Indicate the performance of the study instrument that originated the survey. It is stated to consist of four sections but only three are mentioned. In question 16 there are no options "none" and "others", so the participants may feel obliged to answer any of the indicated options, when perhaps they did not experience it considerably, because in addition the intensity or frequency of the symptom is not measured as to know if the affectation is considerable. What happened to the people who did not answer some questions, were they contacted to complete the survey?
Data analysis.
The use of covariates or confounding variables is mentioned, indicate what they are and their scope in the statistical analysis. It is important that from the beginning of the manuscript the relationship of the independent, of confusion and dependent variables is clear, in order to be clear about what the study consists of, because at the beginning there is no clarity, although in the final sections of the manuscript more is achieved clarity.
There is information in the independent variables section that should be placed in a section on the study instrument.
It states: "These questions were necessary to identify people who would violate lockdown laws intended to prevent the spread of the virus" Is this part of the purpose of the study? What happened to this information? How was it approached from the ethical point of view in relation to the consent of the participants? Also, it is mentioned “These were included because the individuals who perceived the risk are more likely to reduce the spread of the virus.” Is this part of the objective? Is it analyzed or evidenced in the present study?
Better write statistical analysis, each analysis separately. Indicate when statistically significant results are considered, the purpose of each analysis, etc.
Results.
There is a difference between thinking that they can contract the virus and thinking that they can die or be seriously affected, and thinking all of the above does not always indicate feeling distressed, angry, etc. Some results show what is indicated above, but there is no analysis in this regard and it would be important to understand the mental health situation in the participants, that is, there are some “contradictory” or “contrary” results that are valuable and should be analyzed in the discussion, for example in table 2.
Does the option not married, contemplates widowed and divorced?
The likelihood that COVID-19 continues may be influenced by information about the behavior of the virus or by the mass media and not an indicator that shows the type of attitude that the participants have.
Table 3 indicates that the analysis is multivariate and should say univariate.
Remove table 4 after the Discussion section and place it in the Results section, place each table after the paragraph where the table is cited. Indicate the N in each table and indicate when the results are considered statistically significant.
Discussion.
You must keep the "thread" with the other sections. The discussion has very valuable arguments but it is very weak, so it must be rewritten, since there are arguments that are not reflected in the results, others that are not part of the objective of the study and some that do not evaluate in depth, for example the situation of the elderly. It is important to compare the results with similar studies so that it is possible to indicate assessments such as "was higher in this study". There is an argument that can be carried over to the introduction (lines 420-421), as it helps to justify. In summary, there are many results, so the discussion can be richer and more robust, with further analysis of the results, contrasting them with each other and comparing them with other studies, due to their similarities or differences in the evidence and methodology.
Conclusion.
It should be rewritten, since there are arguments that may be debatable or questionable, for example, reviewing whether it is indicated or justified to use expressions such as "high prevalence ..." or "greater risk ..." In addition, they mention that anxiety symptoms are up to three times more frequent ... compared to pre-COVID. Was a comparative analysis carried out or with which study is it corroborated? Therefore, it is necessary to modify the wording or the focus of the conclusion.
Author Response
Response to Reviewer 2 comments and Suggestions
Dear Editor-in-chief,
Thank you for allowing us to revise and re-submit our manuscript. We are incredibly grateful for your insightful feedback that has strengthened our manuscript.
We have addressed all the issues raised by the reviewers in the revised version of the manuscript. Where we have a different opinion from the reviewers, we have clarified our position. All changes made to the revised manuscript have been highlighted in red. Below, we provide a point by point response to your comments and where possible, linked our responses to the appropriate sections in the revised manuscript. We hope that the revisions made have properly addressed your concerns, and that our revised manuscript is acceptable for publication in the journal.
Review of “Prevalence and factors associated with mental health outcomes among Africans during the COVID-19 lockdown period”
|
Reviewer 2: Main comments |
||
|
The study is important, interesting and pioneering, but it requires a reorganization of the sections, in addition to highlighting and deepening even more the arguments related to the objective and results of the study. It is important to follow the style or structure of epidemiological studies. |
The paper has been revised. Sections in the introduction were added, re-written or restructured to aid understanding and to ensure the argument flows logically. |
|
|
Title. Write a more attractive one, perhaps written as a question, and indicate the type of study (considering the variables). |
The title was revised to read: Prevalence and Factors Associated with Mental and Emotional Health Outcomes among Africans during the COVID-19 lockdown period - A web-based cross-sectional study |
|
|
Abstract. For clarity, place the percentages next to each symptom or factor. Write the sentence better “mental health symptoms”, because they do not indicate mental health, (this is repeated a few times in the manuscript, so it must also be modified). |
Symptoms have been used across the manuscript. Percentages have been next to each mental health symptom and the revised section now reads: the prevalence of boredom was 70.5%, followed by feeling anxious (59.1%), worry (57.5%), frustration (51.5%) and anger (22.3%). |
Line 53-54 |
|
Introduction. It must be rewritten. Indicate how similar situations have been addressed in the past and in other countries in the face of the current situation, how the situation has been evaluated in other studies, informing the recommendations of said studies. |
The introduction has been re-written and examples of how similar situation was addressed in the past cited |
Eg. Lines 89-101 |
|
In addition, it is important to define the study variables (they could be noted in a section on the study instrument). |
The study variables have been described in the methods section on survey design and in the data analysis section as well |
|
|
It is important to further develop the argument presented in the penultimate paragraph and the situation of COVID-19 in the African context (paragraph located from line 100 to 106). |
Different sections of the introduction were further developed to present the argument and the situation in the African context |
Lines 102-133 |
|
Write the objective better, in relation to the title and the methodology (considering the variables), which should be developed in the introduction clearly and directly, that is, the relationship between the independent variables and their influence on the dependent variables. |
The aim has been revised in relation to the title. The introduction was also further developed to link the independent variable and the dependent variable. Most of the changes are shown in red throughout the introductory section |
e.gs Lines 134-136
Lines 120-124, 131-133 |
|
Methodology.Study environment and participants. Where did you send the questionnaire when you filled it out? |
Data was collected online via survey monkey by sending e-link of the questionnaires. The completed questionnaires were stored in Survey Monkey regional database and only the anonymized data were retrieved and used for analysis. These has been clarified in the revised manuscript |
Lines 150-151 |
|
Were the answers saved? (This should be indicated in the next section) |
See previous response |
|
|
indicate the characteristics of the participants, as well as the inclusion and exclusion criteria |
Done. A section on inclusion and exclusion criteria was added The characteristics of the respondents are shown in Table 1 |
Section 2.3 |
|
Survey design. Place the survey in an appendix and cite it within the text. |
done |
|
|
Explain in more detail how the study was explained to the participants and the aspects of informed consent. If they did not understand something, did they have any contact to find an explanation? |
The survey preamble had contact email of the lead researcher if participants needed further clarification. This has been included in the revised manuscript The ethics and consent section were revised to reflect the method used during data collection |
Section 2.6 |
|
How were they indicated about anonymity and how was confidentiality assured? (since they are not indicated in the survey) |
This was included in the preamble section of the survey. We have clarified this in the ethics section |
Section 2.6 |
|
How did you control that they were over 18 years of age? |
As this was an online survey, we could only control for this in two ways: As indicated in the revised manuscript, participants were instructed not to complete the survey if they were less than 18 years. Data for those who were aged less than 18 years from response to item 5 of appendix was not included in the analysis. However, this has been added as part of the limitations |
Line 412-414 |
|
In addition, does the study comply with any agreement on an ethical standard? |
The study adhered to the tenets of Helsinki Declaration. The statement has been added in the section on ethics |
Section 2.6 |
|
Some analysis was carried out on the questionnaire, in addition to extracting Cronbach's alpha, to evaluate its suitability? Indicate the performance of the study instrument that originated the survey. |
Agreed and we have modified the text and now reads: The Kudar-Richardson 20 (KR-20) Cronbach’s alpha coefficient measuring internal consistency reliability for measures with dichotomous for the five mental health symptoms ranged from 0.70 to 0.74 indicating satisfactory consistency |
|
|
It is stated to consist of four sections but only three are mentioned. |
Agreed and it has been changed to five sections including listing each section– see line 137-140 |
|
|
In question 16 there are no options "none" and "others", so the participants may feel obliged to answer any of the indicated options, when perhaps they did not experience it considerably, because in addition the intensity or frequency of the symptom is not measured as to know if the affectation is considerable. |
Each of the five COVID-19 lockdown mental and emotional health related measures were binary (yes or no) |
|
|
What happened to the people who did not answer some questions, were they contacted to complete the survey? |
Missing information was not used in our analysis. Additionally, we have provided a footnote in table 2 for clarity which indicates “Total count 2005 unless otherwise given in brackets” |
|
|
Data analysis. The use of covariates or confounding variables is mentioned; indicate what they are and their scope in the statistical analysis. |
We have modified the data analysis section to reflect the confounding variables used |
|
|
It is important that from the beginning of the manuscript the relationship of the independent, of confusion and dependent variables is clear, in order to be clear about what the study consists of, because at the beginning there is no clarity, although in the final sections of the manuscript more is achieved clarity. |
This has been revised across the introduction. See response to previous comment |
|
|
There is information in the independent variables section that should be placed in a section on the study instrument.
It states: "These questions were necessary to identify people who would violate lockdown laws intended to prevent the spread of the virus" Is this part of the purpose of the study? What happened to this information? How was it approached from the ethical point of view in relation to the consent of the participants? |
For clarity and consistency, we have replaced independent variables with confounding variables and based on the journal specification, confounding variables was presented at the right spot. This information was revised to reflect the question. The attitude items were added to examine the potential effect of public compliance to the mitigation practices put inplace to prevent the spread of the virus. This has been revised in the manuscript. The ethics for the project was obtained and participants were informed of the anonymity of responses as detailed in the ethics section of the manuscript in section 2.6 |
|
|
In addition, it is mentioned “These were included because the individuals who perceived the risk are more likely to reduce the spread of the virus.” Is this part of the objective? Is it analyzed or evidenced in the present study? |
Agreed and the whole was re-written for clarity |
|
|
Better write statistical analysis, each analysis separately. Indicate when statistically significant results are considered, the purpose of each analysis, etc. |
Writing each statistical analysis for each outcome variables separately will make the whole statistical section very cumbersome to follow and understand since each of the outcome variables were all binary outcomes. However, we have modified statistical section for more clarity We have indicated the text below which indicated statistically significant results: those with P value < 0.05 were considered as factors associated with the five mental health symptoms (see, bolded adjusted odds and their confidence intervals (CIs) in Table 3). |
|
|
Results. There is a difference between thinking that they can contract the virus and thinking that they can die or be seriously affected, and thinking all of the above does not always indicate feeling distressed, angry, etc. Some results show what is indicated above, but there is no analysis in this regard and it would be important to understand the mental health situation in the participants, that is, there are some “contradictory” or “contrary” results that are valuable and should be analyzed in the discussion, for example in table 2.
|
Done. The data was re-analysed and new tables presented and discussed in the result section. The title on tables 2 was revised |
|
|
Does the option not married, contemplates widowed and divorced? |
Yes. This has been added as footnote |
|
|
The likelihood that COVID-19 continues may be influenced by information about the behavior of the virus or by the mass media and not an indicator that shows the type of attitude that the participants have. |
This was classified as risk perception variables because individuals who perceived the risk will continue are more likely to reduce the spread of the virus. This variable represented item The item 17 of Appendix shows the question: How likely do you think Coronavirus disease (COVID-19) will continue in your country? |
|
|
Table 3 indicates that the analysis is multivariate and should say univariate. |
This was a typo error that we have corrected. The new Table 2 is the univariate while Table 3 is multivariate. |
|
|
Remove table 4 after the Discussion section and place it in the Results section, place each table after the paragraph where the table is cited. Indicate the N in each table and indicate when the results are considered statistically significant. |
Tables have been placed close to where they were first cited. N was included as footnote in Table 1 where the study characteristics were presented. The other tables are odd ratios and adjusted odds ratios and do not need an N. |
|
|
Discussion. You must keep the "thread" with the other sections. The discussion has very valuable arguments but it is very weak, so it must be rewritten, since there are arguments that are not reflected in the results, others that are not part of the objective of the study and some that do not evaluate in depth, for example the situation of the elderly. |
We have modified the discussion for clarity and ensure that it flows to summarise the key findings of the study. |
|
|
It is important to compare the results with similar studies so that it is possible to indicate assessments such as "was higher in this study". |
We also compared our results with other relevant literatures in the subject and have revised the limitation section of the discussion. |
|
|
There is an argument that can be carried over to the introduction (lines 420-421), as it helps to justify. |
Thanks for this. We have reflected similar argument in the introduction but retained it in the discussion because it is inline with what we found and helps to support our finding. |
Lines 128-130 |
|
In summary, there are many results, so the discussion can be richer and more robust, with further analysis of the results, contrasting them with each other and comparing them with other studies, due to their similarities or differences in the evidence and methodology |
We have revised the discussion with reference to previous studies. The tables were also revised to reflect the analysis. The univariate tables (2) was also made clearer. |
|
|
Conclusion. It should be rewritten, since there are arguments that may be debatable or questionable, for example, reviewing whether it is indicated or justified to use expressions such as "high prevalence ..." or "greater risk ...". |
Section was re-written. The use of prevalence was retained as the analysis of prevalence was conducted in the study. |
Lines 408-412 |
|
|
|
|
|
In addition, they mention that anxiety symptoms are up to three times more frequent ... compared to pre-COVID. Was a comparative analysis carried out or with which study is it corroborated? Therefore, it is necessary to modify the wording or the focus of the conclusion |
We have modified the wording in the conclusion and the comparison study has been referenced. |
Lines 410-412 and reference 37 |
Round 2
Reviewer 1 Report
This is a much improved manuscript. Insert "responses" after "dichotomous" in line 168.
Author Response
Reviewers’ comments
Reviewer 1.
Comments and Suggestions for Authors
This is a much improved manuscript. Insert "responses" after "dichotomous" in line 168.
Response: Done
Reviewer 2 Report
Many congratulations for the effort made, the result has been very good. The discussion is very important and necessary. I hope this study will have a great diffusion.
The missing changes are minimal but essential for the article to be more orderly and clearly:
It is not necessary to put statistical results in the summary (line 60).
I suggest adding as keywords: boredom and anger.
Record the Cronbach's Alpha from the original survey (reference 21).
Place Appendix A when it is mentioned in the text (sections: Survey design and data analysis) and indicate in the Appendix A section: Appendix A1. Questionnaire…
Move the Consent and ethical considerations section as section 2.4, after Inclusion and exclusion criteria and before Data analysis.
It is important to define the dependent variables synthetically avoid confusion between terms, for example anxiety and worry.
Place Figure 1 after section 3.2 Prevalence of mental health / emotional symptoms.
The only data that requires clarification is:
It is mentioned that individuals who reported that they did not feel concerned that they or their family members would be infected with COVID-19 had the lowest probabilities in all variables… but it is associated with “… with higher odds of mental health / emotional symptoms in this study…” It is a writing problem that does not allow greater clarity and generates confusion because it sounds contradictory. I suggest you write it apart from the other results, that is, that the results that indicate a greater presence of symptoms are differentiated... and those that indicate the opposite not put them together in a single sentence.
The foregoing is indicated by the information provided in the summary and lines 267-270, 296-300, 315-318.
Author Response
Reviewer 2
Many congratulations for the effort made, the result has been very good. The discussion is very important and necessary. I hope this study will have a great diffusion.
The missing changes are minimal but essential for the article to be more orderly and clearly:
It is not necessary to put statistical results in the summary (line 60).
Response: Statistical result was deleted
I suggest adding as keywords: boredom and anger.
Response: We have added bored and angry to the keywords in line with the study outcomes.
Record the Cronbach's Alpha from the original survey (reference 21).
Response: Done. The sentence was revised to include the Cronbach alpha score. It now reads: The survey instrument was adapted and developed from WHO-recommended questions used in a previous survey (Cronbach’s alpha coefficients of 0.74)….
Place Appendix A when it is mentioned in the text (sections: Survey design and data analysis) and indicate in the Appendix A section: Appendix A1. Questionnaire…
Response: Done: This self-administered online questionnaire shown in Appendix A.
The Appendix A heading was also revised.
Move the Consent and ethical considerations section as section 2.4, after Inclusion and exclusion criteria and before Data analysis.
Response: Done
It is important to define the dependent variables synthetically avoid confusion between terms, for example anxiety and worry.
Response: This was revised to read: The outcome variables in this study was derived from the item asking participants: ‘how do you feel about the COVID-19 lockdown measures?’ (Item 16 of Appendix A). The measures were ‘frustrated’, ‘angry’, ‘bored’, ‘anxious’ and ‘worried’ about COVID-19. Each of these five outcome variables were coded as binary, ‘1’ for yes and ‘0’ for no.
Place Figure 1 after section 3.2 Prevalence of mental health / emotional symptoms.
Response: Done
The only data that requires clarification is:
It is mentioned that individuals who reported that they did not feel concerned that they or their family members would be infected with COVID-19 had the lowest probabilities in all variables… but it is associated with “… with higher odds of mental health / emotional symptoms in this study…” It is a writing problem that does not allow greater clarity and generates confusion because it sounds contradictory. I suggest you write it apart from the other results, that is, that the results that indicate a greater presence of symptoms are differentiated... and those that indicate the opposite not put them together in a single sentence.
The foregoing is indicated by the information provided in the summary and lines 267-270, 296-300, 315-318.
Response: Done:
Lines 267-270 was revised to: Individuals who were concerned that they or their family members could be infected with COVID-19 were less likely to feel worried and less anxious about contracting the infection. Similarly, participants who felt at lower risk of infection (OR 0.34, 95%CI 0.27, 0.41), severe infection (OR 0.26, 95%CI 0.20, 0.33) and those who thought their risk of dying from the infection was low (OR 0.18, 95%CI 0.14, 0.25), were less likely to worry about COVID-19 in this study.
296-300 was revised to: Overall, respondents who perceived a low risk of being infected by COVID-19 were less likely to be worried about the disease and those from Southern African were less likely to feel bored during the pandemic (aOR 0.59, 95%CI 0.42, 0.82), compared to other respondents.
315-318 was revised to: In addition, respondents who felt that their risk of being infected with the virus was low and those who do not think that COVID-19 will continue after the lockdown were more likely to feel angry about the pandemic. The study also found that people working in health care sectors were less likely to report mental and emotional health symptoms during the COVID-19 pandemic.